# Harnessing enzyme promiscuity of alditol-2-dehydrogenases for oxidation of alditols to enantiopure ketoses

Prithwiraj De[1]*, Jenna Salvat[1,¤a], Eliza Walthers[1], James Henriksen[1], Michael Wells[1,¤b], Richard T. Conant[1], Claudia M. Boot[1,2]

1 Natural Resource Ecology Laboratory, Colorado State University, Fort Collins, Colorado, United States of America, 2 Department of Chemistry, Colorado State University, Fort Collins, Colorado, United States of America.

¤a Current address: Geology and Geological Engineering, Colorado School of Mines, Golden, Colorado, United States of America
¤b Current address: School of Biological Sciences, Louisiana Tech University, Ruston, Louisiana, United States of America
* prithwiraj.de@colostate.edu, prithwiraj.de@gmail.com

## Abstract

Several alditol-2-dehydrogenase enzymes from the short chain dehydrogenase (SDR) family catalyze the production of enantiopure rare ketoses from alditols as substrates and are used in biotech industry. Clearly, the absolute configuration of the internal chiral carbons in the open-chain conformation of alditols provides the structural basis for the enzymatic operation. This also allows for substrate ambiguity that manifest as enzyme promiscuity due to partial stereoselectivity of the enzyme. The issue is to make the right choice of a promiscuous enzyme and access maximum product diversity. Based on the absolute configuration of a cohort of ten enantiopure hexitols, this study is a systematic exploration of the stereochemical foundation of enzyme promiscuity in alditol-2-dehydrogenases which does not involve any kinetic analysis. Using cell-free expressed galactitol-2-dehydrogenase (G2DH), D-sorbitol-2-dehydrogenase (D-S2DH), and D-altritol-5-dehydrogenase (D-A5DH), we confirmed chemoenzymatic synthesis of all enantiopure ketohexoses through characterization by GC/MS and NMR spectroscopy. However, we found that enzyme promiscuity is beyond the partial stereoselectivity for certain alditol-2-dehydrogenase enzymes that are reliably producing enantiopure ketoses from multiple alditols. For instance, G2DH oxidizes galactitol (2R 3S; 55% conversion), a *meso*-alditol with a plane of symmetry, as well as L-talitol (2S 3S; 2.1% conversion) to L-tagatose. The substrate bears different absolute/relative configurations and is an example among several promiscuous oxidations. Notably, their product yields are different under similar reaction conditions indicating stereochemical preference of the enzyme. This *in vitro* chemoenzymatic investigation of the underlying stereochemical interplay for the enantioselective oxidation of alditols explores the potential to harness enzyme promiscuity/

**Data availability statement:** All relevant data are within the paper and its Supporting Information files.

**Funding:** Funding for this research was supported by the Keck Research Foundation, the Grantham Foundation, and Grant #2000-67030-31475 from the US Department of Agriculture

substrate-adaptability as a tool for the predictive synthesis of multiple enantiopure ketoses. This study focuses on unconventional stereochemical investigation into enzyme stereospecificity and promiscuity which precludes kinetic analysis. Given the role of enzyme promiscuity in evolutionary processes, systematic stereochemical analysis may prove crucial in future.

## Introduction

A typical monosaccharide is stereochemically classified as either D- or L-configuration based on the orientation of the bottom-most chiral center in its Fischer projection and implies its relationship to its enantiomeric mirror image. The D-enantiomer is generally more prevalent in nature and has been extensively studied in chemical, biochemical, and biotechnological contexts [1,2]. In contrast, certain enantiomers of rare sugars and alditols are either absent in nature or exist in extremely low abundance [3]. All L-hexoses, by definition, are considered rare sugars and are notably expensive [3]. Their limited availability and high costs have hindered extensive research and practical applications.

Rare ketohexoses and alditols (Fig 1) have gained significant interest as targets for commercial production due to their potential uses as alternative sweeteners, pharmaceutical agents (e.g., anti-obesity drugs, tumor suppressants, antidiabetic medications), and herbicides or insecticides [4–7]. There are four common ketohexoses (Fig 1); psicose (**1a,b**), fructose (**2a,b**), tagatose (**3a,b**) and sorbose (**4a,b**) and each exists in two enantiomeric (nonsuperimposable mirror image) forms. The D-enantiomers of two rare hexoses, D-psicose (**1a**; also known as D-allulose) and D-tagatose (**3a**), have been certified by the FDA as "generally recognized as safe" and are now produced on a commercial scale [8,9]. Similarly, L-sorbose (**4b**) has been employed for decades as an intermediate in the industrial synthesis of L-ascorbic acid (vitamin C) [10]. L-tagatose (**3b**) serves as a precursor for the synthesis of deoxygalactonojirimycin (DGJ), a pharmacological chaperone with potential applications in treating lysosomal storage disorders [11]. Furthermore, L-fructose (**2b**) shows promise as a nonnutritive sweetener [12] and an effective inhibitor of glycosidases [13]. Our goal was to establish a stereochemical framework for selecting alditol-2-dehydrogenase in the synthesis of all enantiopure of ketohexoses. Therefore, it was essential to thoroughly investigate the molecular mechanism underlying current methods that employ alditol-2-dehydrogenase-catalyzed oxidation of alditols.

### Biotechnological production and the Izumoring hypothesis

General methodologies for the synthesis of L-hexoses are being explored in recent times, with the Izumoring hypothesis emerging as the preferred approach in the biotechnology industry [3,14–16]. Introduced over two decades ago, the Izumoring hypothesis putatively guides the bioproduction of all hexose sugars, including 16x aldohexoses, 8x ketohexoses, and 10x hexitols, through enzymatic and microbiological reactions [16]. A central concept of this hypothesis involves the use of polyol

## Ketohexoses

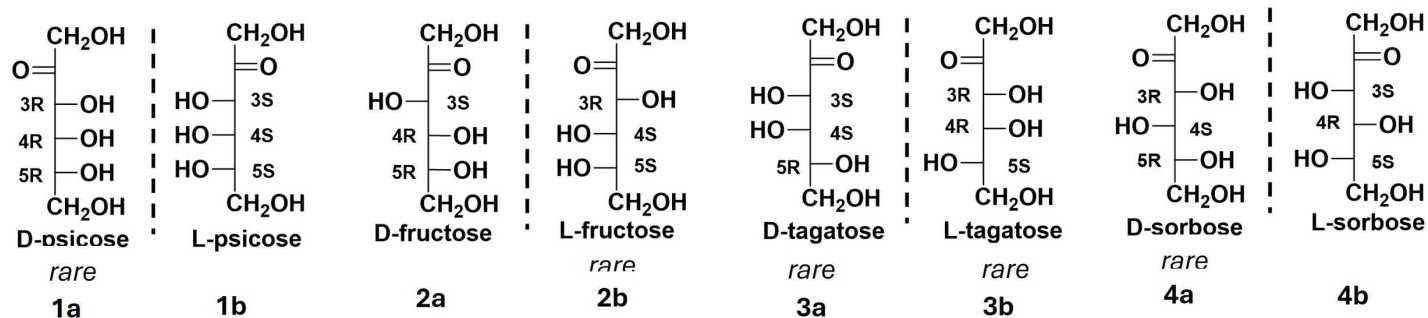

## Hexitols

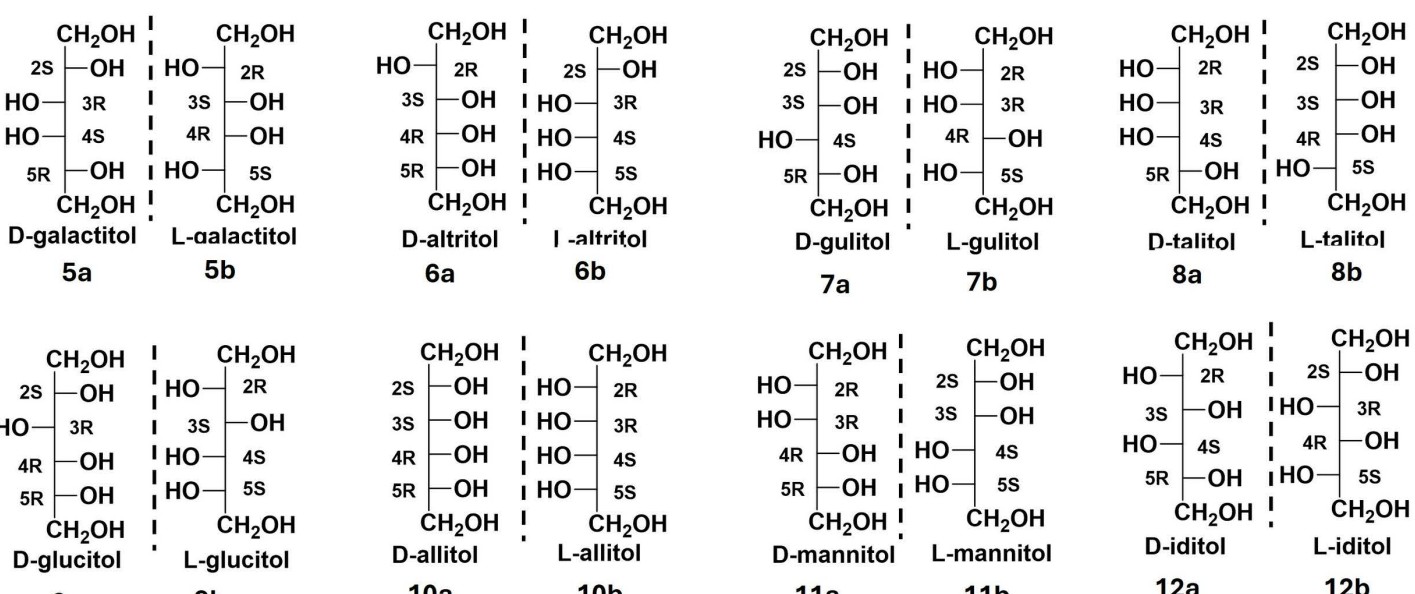

**Fig 1. Ketohexoses and hexitols with their enantiomers (mirror images).** The orientation of the hydroxyl (-OH) to right (D-) and left (L-) determines the enantiomeric identities.

(alditol) dehydrogenases, which catalyze oxidation-reduction reactions between ketohexoses and their corresponding hexitols. In the Izumoring hypothesis, poly-ol dehydrogenases have been classified based on their substrate specificities, particularly their recognition of hydroxyl (-OH) group (relative) configurations at the C2 and C3 positions of alditols. This was based upon the studies on *Bacillus pallidus* that revealed the specificity of pentitol dehydrogenases (e.g., ribitol 2-dehydrogenase, ribitol 4-dehydrogenase, xylitol-2-dehydrogenase, and xylitol 4-dehydrogenase) for D-*erythro* and D-*threo* configurations [17]. These enzymes belong to the short-chain dehydrogenase-reductase (SDR) superfamily [18,19], which produces ketoses by oxidizing the 2-position of alditols (Fig 2). The concept is based in the relative configuration (D-/L-*erythro/threo*) and not on the absolute configuration of the hydroxyl groups at C2 and C3 positions. Interestingly, if an enzyme recognizes only the stereochemistry at C2 and C3 positions of a substrate polyol, it may catalyze reactions involving different alditols with similar configurations at these positions but different at other chiral centers. For

| A | | B | | C | | D | |
|---|---|---|---|---|---|---|---|
| **D-erythro** (C2-C3) **Substrate** | Ketose product | **L-erythro** (C2-C3) **Substrate** | Ketose product | **L-threo** (C2-C3) **Substrate** | Ketose product | **D-threo** (C2-C3) **Substrate** | Ketose product |
| D-allitol (**6a**) § 2S 3S 4R 5R | D-psicose (**1a**) 3R 4R 5R | D-mannitol (**11a**) 2R 3R 4R 5R | D-fructose (**2a**) 3S 4R 5R | D-glucitol (**9a**) 2S 3R 4R 5R | D-fructose (**2a**) 3S 4R 5R | D-talitol (**8a**)¥ 2R 3R 4S 5R | D-psicose (**1a**) 3R 4R 5R |
| L-mannitol (**11b**) 2S 3S 4S 5S | L-fructose (**2b**) 3R 4S 5S | L-allitol (**6b**) § 2R 3R 4S 5S | L-psicose (**1b**) 3S 4S 5S | L-talitol (**8b**)¥ 2S 3S 4R 5S | L-psicose (**1b**) 3S 4S 5S | L-glucitol (**9b**) 2R 3S 4S 5S | L-fructose (**2b**) 3R 4S 5S |
| L-talitol (**8b**) 2S 3S 4R 5S | L-tagatose (**3b**) 3R 4R 5S | D-glucitol (**9a**)¥ 2S 3R 4R 5R | L-sorbose (**4b**) 3S 4R 5S | L-iditol (**12b**) 2S 3R 4R 5S | L-sorbose (**4b**) 3S 4R 5S | L-galactitol (**5b**) § 2R 3S 4R 5S | L-tagatose (**3b**) 3R 4R 5S |
| L-glucitol (**9b**)¥ 2R 3S 4S 5S | D-sorbose (**4a**) 3R 4S 5R | D-talitol (**8a**) 2R 3R 4S 5R | D-tagatose (**3a**) 3S 4R 5R | D-galactitol (**5a**)§ 2S 3R 4S 5R | D-tagatose (**3a**) 3S 4S 5R | D-iditol (**12a**) 2R 3S 4S 5R | D-sorbose (**4a**) 3R 4S 5R |
| L-ribitol (**13b**) §¥ 2S 3S 4R | D-ribulose (**14a**) 3R 4R | **D-arabitol** (**15a**)¥ **2R 3 4R** | D-xylulose (**16a**) 3S 4R | **D-xylitol** (**17a**)§ **2S 3 4R** | D-xylulose (**16a**) 3S 4R | D-arabitol (**15a**) 2R 3S 4R | D-ribulose(**13a**) 3R 4R |
| **L-arabitol** (**15b**)¥ **2S 3 4S** | L-xylulose (**16b**) 3R 4S | **L-ribitol** (**13b**) § **2S 3S 4R** | L-ribulose (**14b**) 3S 4S | **L-arabitol** (**15b**) **2S 3 4S** | L-ribulose (**14b**) 3S 4S | **L-xylitol** (**17b**) § **2R 3R 4S** | L-xylulose (**16b**) 3R 4S |
| **D-erythritol** (**18a**) § **2S 3R** | D-erythrulose (**19a**) 3R | **L-erythritol** (**18b**) § **2R 3S** | L-erythrulose (**19b**) 3S | **L-threitol** (**20a**) **2S 3S** | L-erythrulose (**21b**) 3S | **D-threitol** (**20b**) **2R 3R** | D-erythrulose (**21a**) 3R |
| 2S 3S System for hexitols | | 2R 3R system for hexitols | | 2S 3R system for hexitols | | 2R 3S system for hexitols | |
| § *meso*; other enantiomer should also produce same product ketose; ¥ approach of enzyme: C5-C4(for hexitols)/C4-C3 (for pentitols) | | | | | | | |

**Fig 2. The alditol to ketose conversion, catalyzed by respective alditol-2-dehydrogenases, table [17] is modified with the introduction of absolute configuration of alditols and corresponding ketoses: The four columns of alditols represent 2S 3S/ D-*erythro* (A), 2R 3R/ L-*erythro* (B), 2S 3R/ L-*threo* (C) and 2R 3S/ D-*threo* (D) systems respectively for the hexitol category (Blue, C2; Red, C3 in substrate alditol).** It is hypothesized that one enzyme, specific to one absolute configuration system, should be able to catalyze the conversion of all the alditols listed in the same column. Alditols in bold italics do not match absolute configurations of hexitols listed in the same column.

example, if the enzyme is partially selective for D-*threo* / 2R 3S configuration, this enzyme should also catalyze oxidation of both alditols having 2R 3S 4R and 2R 3S 4S configurations. Therefore, the alditol-2-dehydrogenase enzymes are qualified as promiscuous enzymes [20,21].

Enzymatic promiscuity arises when one enzyme acts on multiple structurally related substrates. Enzyme promiscuity has been linked directly to evolution and plenty of work has been published regarding enzyme/protein architecture to discover underlying structural sources of specificity and substrate ambiguity [22]. Beyond catalyzing single metabolic reactions, certain enzymes perform broader cellular functions such as proofreading, nutrient scavenging, antimetabolite removal, metabolite pool balancing, and system redundancy. Notably, these roles are exemplified by the phosphatases of the haloalkanoate dehalogenase superfamily and the thioesterases of the hotdog fold superfamily [22]. Currently, little is known about the stereochemical parameters governing enzyme promiscuity of, particularly with respect to substrate scope and strategies to harness this phenomenon for synthetic advantage. Our primary goal was to exploit enzyme promiscuity

to establish an approach to one enzyme-one protocol-multiple enantiopure ketohexose product platform. However, we found that predicting enzyme stereoselectivity remains challenging; especially when the substrates exist solely in open-chain conformation, as in the case of polyols, where the enzyme may approach from either end of the molecule. This explains why the commercial exploitation of the Izumoring hypothesis has been limited by the lack of stereochemical generality of the substrates towards different alditol dehydrogenases, thereby creating unpredictability of their substrate specificities [23]. We aimed to investigate this issue from the standpoint of absolute configuration of the substrates. The present study explores this hypothesis using a model stereochemical group of 10-hexitols. This study also provides systematic experimental evidence to support the hypothesis that enzyme promiscuity may stem from the enzyme's partial selectivity toward similar stereo- and regio-chemistry of substrates, rather than entire structure of the molecule. This concept has been utilized to synthesize enantiopure rare ketoses in the present work.

## Results

We prepared three hexitol-2-dehydrogenases (Fig 3) using cell-free expression systems; 1) Galactitol-2-dehydrogenase (G2DH) from a mutant strain of *Rhodobacter spheroides*, known to transform galactitol (**5a,b**) into L-tagatose (**3b**) [24–26]. This shows that the enzyme is specific for D-*threo* / 2R 3S relative/absolute configuration of C2-C3 of L-galactitol (**5b**). 2) D-altritol-5-dehydrogenase (D-A5DH) from *Agrobacterium tumefaciens c58*, which catalyzes the conversion of D-altritol

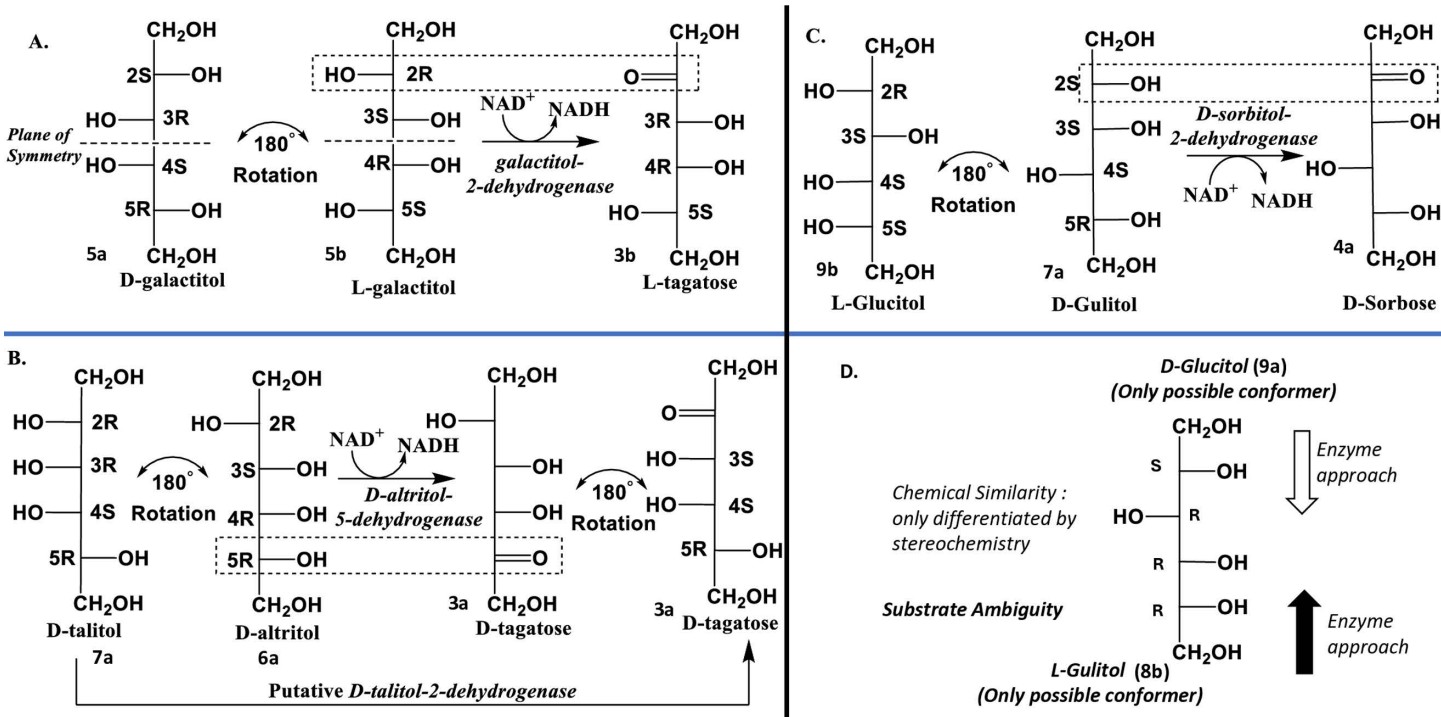

**Fig 3. The substrate specificities of three alditol −2-dehydrogenases in terms of the absolute configuration of C2 and C3; A, G2DH has specificity for 2R 3S systems:** L-tagatose (3b) preserves the stereochemistry of C3, C4 and C5 of L-galactitol (5b), a *meso* compound as it is superimposable with D-galactitol (5a) after 180° rotation in plane. **B, D-A5DH has specificity for 2R 3R systems:** D-altritol (6a) produces D-tagatose (3a) after oxidation at C5; also, D-altritol (6a) attains configurations of D-talitol (7a) after 180° rotation in plane and a putative D-talitol-2-dehydrogenase should produce D-tagatose (3a). **C, S2DH has specificities for 2S 3S systems:** D-gulitol (8a) (i.e., L-glucitol (9b) after 180° rotation in plane) getting oxidized at C2 to give D-sorbose (4a). **D,** Functional group similarities from both ends of alditols may, hypothetically, offer substrate ambiguity to alditol dehydrogenases. For example, alditol-2-dehydrogenase may approach D-glucitol (9a) from top and L-gulitol (8b) from bottom.

(6a) into D-tagatose (3a) [27] showing L-*erythro* / 2R 3R specific activity and 3) D-sorbitol-2-dehydrogenase (D-S2DH) from *Bradyrhizobium japonicum*, which oxidizes D-sorbitol (D-gulitol (7a)) to D-sorbose (4a) [28–30] showing D-*erythro* / 2S 3S configuration specific activity. These enzymes have never been investigated for their promiscuous role.

Notably, the oxidation of alditols by these enzymes preserves the absolute configuration of all other chiral centers. For example, the *meso* form of galactitol (D- or L-; 5a,b) can be interconverted *via* 180° planar rotation, making both enantiomers suitable substrates for G2DH. The other example is that D-A5DH is practically oxidizing 2-position of D-talitol (8a) (i.e., D-altritol (6a) *via* 180° planar rotation). Therefore, D-A5DH can also be named D-talitol-2-dehydrogenase (D-T2DH). Similarly, L-glucitol (9b) attains the configuration of D-gulitol (7a) *via* 180° planar rotation. Therefore, the structural ambiguity of hexitols, due to their identical -CH$_2$OH groups at both ends, open chain conformation and 180° planar rotation, complicate substrate recognition and predictive assessment of enzymatic stereospecificity.

## Resolving stereochemical challenges

We applied the Cahn-Ingold-Prelog (CIP) priority rules to assign absolute R/S configurations to the chiral centers of alditols presented in the modified Izumoring table (Fig 2). This was to provide a universal and precise stereochemical framework. Alditols were numbered based on their parental aldoses, facilitating consistent structural interpretations. The incorporation of absolute configurations of chiral centers in alditols revealed a potential correlation among hexitols in the same column. Hexitols within each column share identical absolute configurations (R/S) for C2 and C3 positions. Consequently, each column a/b/c/ and d represents distinct relative configurations for C2-C3 position and hexitols also present systems: 2S 3S, 2R 3R, 2S 3R, and 2R 3S, respectively. However, pentitols (13a,b-17a,b) and tetritols (18a,b-21a,b) occasionally deviate from this absolute configuration nomenclature, though they remain viable substrates based on relative configurations.

## Categorization of alditols

The hexitols were classified into four structural groups relevant to SDR-catalyzed oxidation (Fig. 3): **Group I:** Alditols with a plane of symmetry through the C3–C4 bond. Examples include galactitol (5a, 5b) and allitol (10a,b; Fig 1). Upon 180° rotation (Fischer projection), these compounds form superimposable enantiomers (D-/L-forms). **Group II:** Mannitol (11a,b; Fig 1) and iditol (12a,b; Fig 1) maintain their enantiomeric identities after 180° rotation. For instance, D-mannitol (11a) remains D-mannitol, and D-iditol (12a) remains D-iditol. **Group III:** Altritol (6a,b) and talitol (8a,b) interchange configurations upon 180° rotation but preserve enantiomeric identities. For instance, L-altritol (6b) adopts the configuration of L-talitol (8b) and *vice versa* after rotation. **Group IV:** Glucitol (9a,b) and gulitol (7a,b) interchange both configurations and enantiomeric identities. For example, D-glucitol (9a) assumes the configuration of L-gulitol (7b) and *vice versa* after rotation.

For our study, we chose galactitol (5a,b) and allitol (10a,b), both '*meso*', from Group II, both D and L-mannitol (11a,b) and D-/L-iditol (12a,b) from Group II, D-and L-talitol (8a,b) from Group III and D- & L-glucitol (9a,b) from Group IV (Table 1). With these 10 alditols, we could include all possible stereochemical combinations of C2 and C3 (i.e., 2S 3S, 2R 3R, 2S 3R and 2R 3S system alditols representing each quarter (Fig 4). Notably, the absolute configuration analysis ensures that alditols from each group satisfies all possible R/S combinations for C2 and C3. Groups I and III exhibit opposite configurations (RS/SR) at C3 and C4, while Groups II and IV maintain similar configurations (RR/SS). We also tested 2-$^{13}$C-L-glucitol (9'b; Fig 5) 2-$^{13}$C-D-xylitol (17'a; Fig 6) as substrates for G2DH.

## Analytical techniques and observations

The product ketoses were subjected to methoxime-TMS derivatization [31] and were quantitatively detected by GC/MS. L-xylulose (14b, Fig 6) was used as the internal standard. Chromatograms revealed distinct peaks corresponding to tautomers (all chromatograms S1 Fig), with one unique peak used for quantitation. The tautomeric ratio is expected to be

**Table 1. The alditol substrate cohort and corresponding product ketohexoses (% yield), determined by GC/MS, showing alditol-2-dehydrogenase activities.** The enzymes being partially stereospecific, able to show promiscuity over different substrates with similar partial absolute configuration.

| Group | Substrates taken | Substrate after 180º rotation in plane | Product with G2DH (2R 3S specific) (yield %) | Product with A5DH (2R 3R specific) (yield %) | Product with S2DH (2S 3S specific) (yield %) |
|---|---|---|---|---|---|
| I | D-galactitol; 5a 2S 3S 4S 5R | L-galactitol; 5b 2R 3S 4S 5S | L-tagatose; 3b (55%) 3R 4R 5S | No ketose product | No ketose product |
| | D-allitol; 10a 2S 3S 4R 5R | L-allitol; 10b 2R 3R 4S 5S | L-psicose*; 1b (2%) 3S 4S 5S | L-psicose; 1b (0.29%)$ 3S 4S 5S | No ketose product |
| II | D-mannitol; 11a 2R 3R 4R 5R | D-mannitol; 11a 2R 3R 4R 5R | No ketose product | D-fructose; 2a (0.63%) 3S 4R 5R | No Ketose product |
| | L-mannitol; 11b 2S 3S 4S 5S | L-mannitol; 11b 2S 3S 4S 5S | No ketose product | No ketose product | No ketose product |
| | D-iditol; 12a 2R 3S 4S 5R | D-iditol; 12a 2R 3S 4S 5R | D-sorbose; 4a (17%) 3R 4S 5R | No ketose product | No ketose product |
| | L-iditol; 12b 2S 3R 4R 5S | L-iditol; 12b 2S 3R 4R 5S | No ketose product | No ketose product | No ketose product |
| III | D-altritol; 6a 2R 3S 4R 5R | D-talitol; 7a 2R 3R 4S 5R | D-psicose; 1a (11%) 3R 4R 5R + D-tagatose; 3a (3.2%) 3S 4S 5R* | D-psicose; 1a (0.37%) 3R 4R 5R*$ + D-tagatose; 3a (0.63%) 3S 4S 5R | No ketose product |
| | L-altritol; 6b 2S 3R 4S 5S | L-talitol; 7b 2S 3S 4R 5S | L-psicose; 1b (1.9%) 3S 4S 5S* + L-tagatose; 3b (2.1%) 3R 4R 5S* | L-psicose; 1b (0.65%) 3S 4S 5S* + L-tagatose; 3b (0.74%) 3R 4R 5S* | L-tagatose; 3b (0.47%) 3R 4R 5S |
| IV | D-glucitol; 9a 2S 3R 4R 5R | L-gulitol; 7b 2R 3R 4R 5S | L-sorbose; 4b (1.6%) 3S 4R 5S | No ketose product | No ketose product |
| | L-glucitol; 9b 2R 3S 4S 5S | D-gulitol; 7a 2S 3S 4S 5R | D-fructose; 2b (18.5%) 3R 4S 5S | No ketose product | D-sorbose; 4a (1.15%) 3R 4S 5R |

*Stereochemistry of C2-C3 did not match $ enantiomer was predicted, could not be determined (low yield).

constant in the same solvent and therefore, only one unique peak may be considered as the quantitation peak in GC/MS (S2 Fig). Notably, overlapping peaks among ketoses (e.g., tagatose (**3a,b**), psicose (**1a,b**), sorbose (**4a,b**), and fructose (**2a,b**)) necessitated the use of unique derivatives and chiral columns for separation.

Trimethylsilylation of ketoses in pyridine derivatize different conformers (such as α/β-pyranose/furanose) and therefore, silylated derivatives are diastereomeric in nature. A β-cyclodextrin based chiral column was used to resolve silylated ketoses by GC/MS (For example L-xylulose **14b** in Fig 6). Chromatograms for the rest of the product ketoses are presented with S4 Fig.

In selected reactions, $^{13}$C NMR spectroscopy was used for $^{13}$C2 of labeled alditols and enantiomers (also purity) were identified. Though NMR spectroscopy is grossly less sensitive than mass spectroscopy, the labeling (2-$^{13}$C) allowed us to monitor changes happening on and around C-2 of the alditol even at comparatively low concentration as per NMR requirements of natural compounds.

## Reaction with galactitol-2-dehydrogenase

Under the optimized reaction conditions, galactitol (**5a,b**; Group I) produced L-tagatose (**3b**) in 55% yield (Table 1) with G2DH catalysis. In fact, all the hexitols that have a 2R 3S absolute configuration or D-*threo* relative configuration, such as

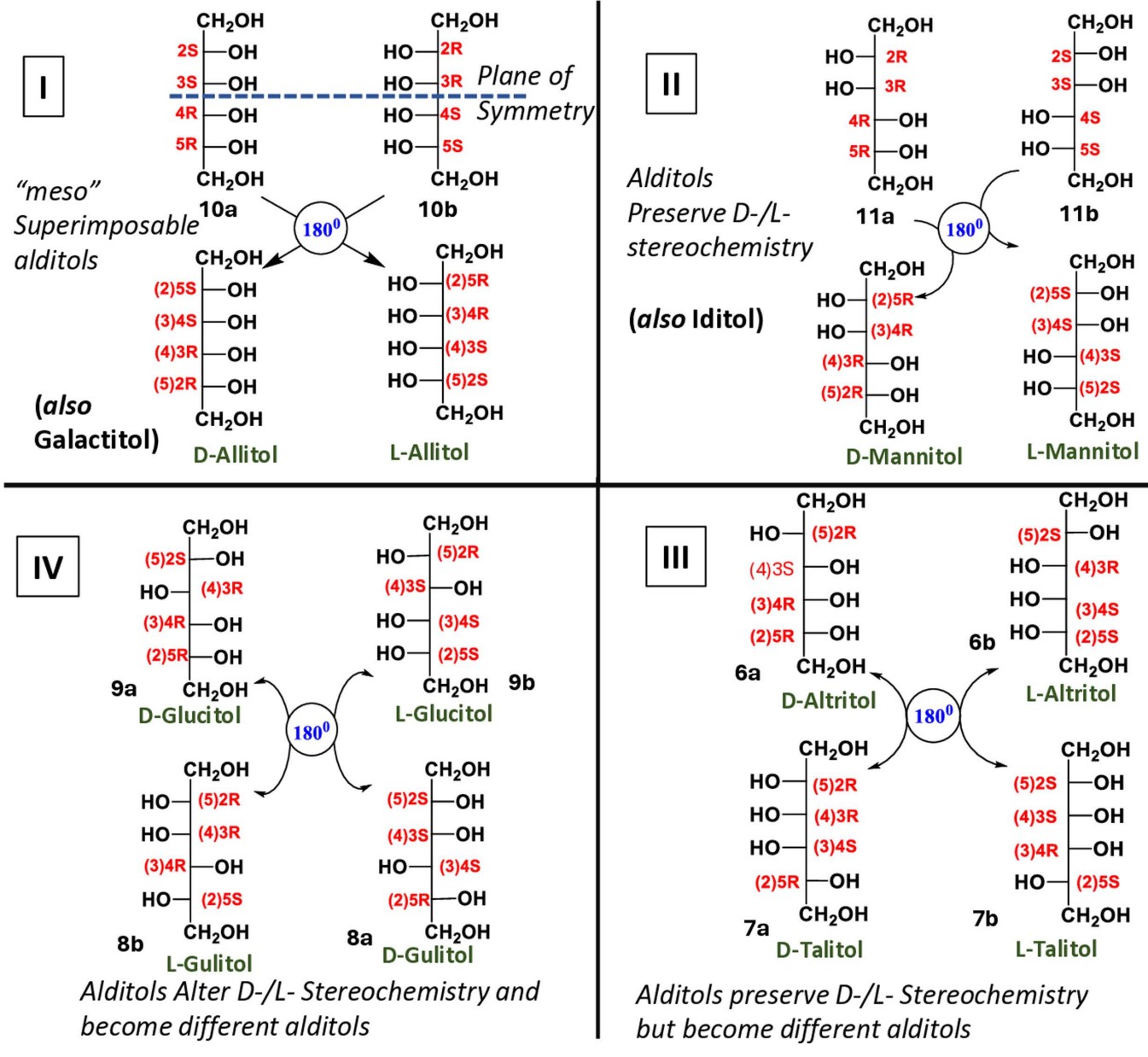

**Fig 4. Examples of the four categories of stereochemical relationships of D- & L- hexitols after 180° rotation in plane:** Altered absolute configurations of the chiral centers and their numbering from the parent aldohexose: Clockwise; Group I include galactitol (5a,b) and allitol (10a,b) both are *meso* compounds as D-and L-enantiomers are superimposable. Group II includes iditol (12a,b) and mannitol (11a,b) (both D-and L-enantiomers); preserves respective enantiomeric identities(D-/L-). Group III includes altritol (6a,b) and talitol (8a,b); these alditols preserve enantiomeric identities (D-/L-) but become new alditols. D-glucitol (9a) becomes L-gulitol (8b) and L-glucitol (9b) becomes D-gulitol (8a) and *vice versa*; included in **Group IV**.

D-iditol (2R 3S 4S 5R; Group II, **12a**) produced D-sorbose (**4a**) (17%), D-talitol (**8a**) (D-altritol (**6a**) (2R 3S 4R 5R; Group III; after 180º rotation in plane) produced a mixture of D-psicose (11%; **1a**) and D-tagatose (**3a**) (3.2%) and L-glucitol (**9b**) (2R 3S 4S 5S; Group IV) produced L-fructose (18.5%; **2b**) respectively by G2DH catalysis. The peak at δ 73.0 ppm (C2) was obtained for 2-$^{13}$C-L-glucitol (**9'b, Fig 5A-C**). The reaction mixture of 2-$^{13}$C-L-glucitol (**9'b**) with G2DH showed three

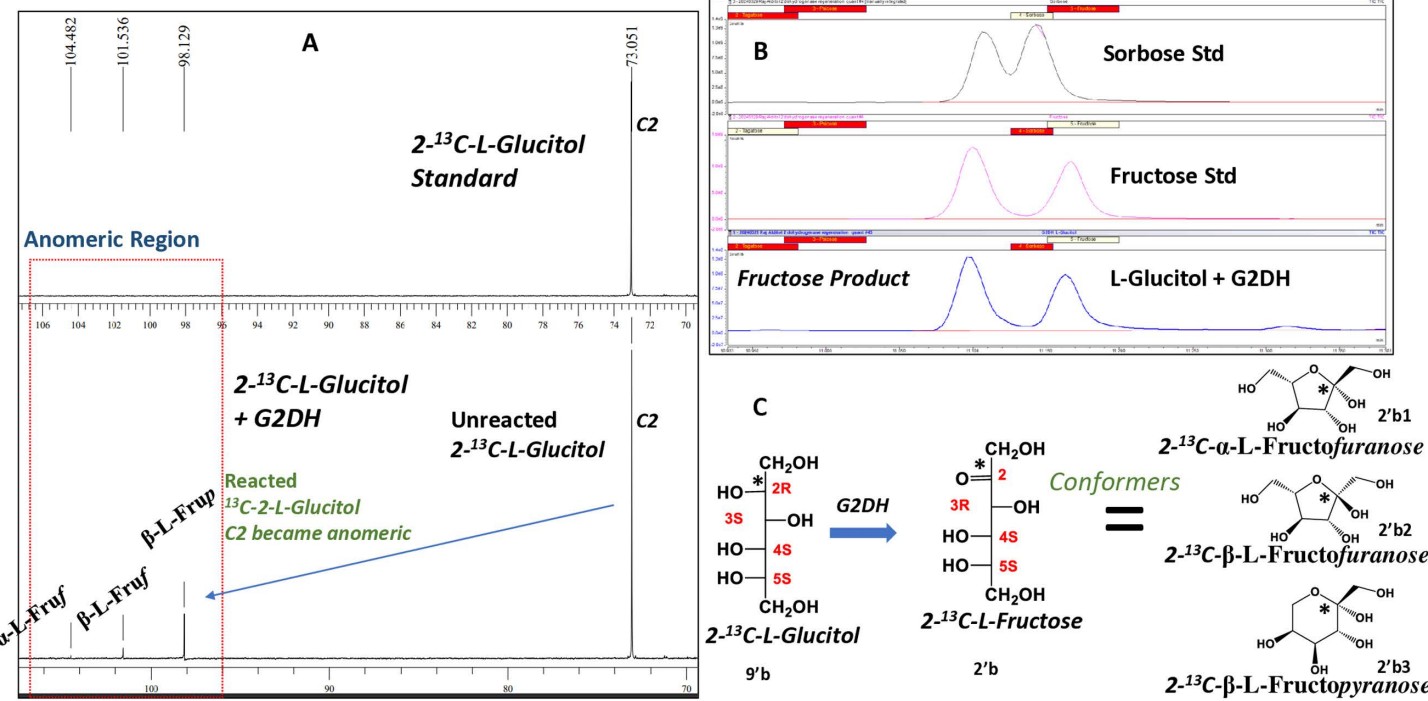

**Fig 5. Molecular characterization of product 2-¹³C-L-fructose (22) from the reaction of 2-¹³C-L-glucitol (21) in the presence of galactitol-2-dehydrogenase: A**, proton decoupled ¹³C NMR spectrum of 2-¹³C-L-glucitol (**9′b**) standard (top panel) showing signal of C2 at δ 73.0 ppm and reaction mixture of 2-¹³C-L-glucitol (**9′b**) in the presence of G2DH (bottom panel); Three new peaks at δ 98.1, δ 101.5 and δ 104.4 ppm at the anomeric range were attributed to three dominant conformers of 2-¹³C-L-fructose (**2′b**). **B**, Methoxime-TMS derivatives detected by GC/MS confirmed fructose as the product: Sorbose standard (top panel), fructose standard (middle panel) and reaction mixture (bottom panel). **C**, Reaction scheme showing oxidation of C2 and preservation of enantiomeric identity of 2-¹³C-L-glucitol (**9′b**) producing 2-¹³C-L-fructose (**2′b**), the enantioselective product, by G2DH.

additional peaks at δ 104.4, δ 101.5 and δ 98.1 ppm in addition to the peak at δ 72.0 ppm (Fig 5A). Since the chemical shifts of the new peaks were found in the anomeric region, the peaks assigned as conformers of C2 of a ketose (**22a-c**). The GC/MS detection of fructose supports the assignment (Fig 5B). These clearly indicate that the G2DH oxidation of L-glucitol (**9b**) produced L-fructose (**2b**) since the configuration of C3, C4 and C5, as seen in Fischer projection formula, should remain unaltered while the designation of absolute configuration may change (Fig 5C). Surprisingly, L-allitol (**10b**, *meso*, 2R 3R 4S 5S; Group I) produced L-psicose (2%; **1b**), L-talitol (**8b**) (2S 3S 4R 5S; Group III) and L-altritol (**6b**) (3S 3R 4S 5S; Group III) produced a mixture of L-tagatose (**3b**) (2.1%) and L-psicose (1.9%; **1b**), L-gulitol (**7b**) (2R 3R 4R 5S) Group IV, were also oxidized to L-sorbose (**4b**) (1.6%) by G2DH catalysis.

In an investigation, 2-¹³C-D-xylitol (**17′a;** Fig 6), a representative pentitol, did not produce 2-¹³C-D-xylylose (**14a**). Instead, **13a** produced 4-¹³C- L-xylulose (**14b**), a keto pentose, due to the oxidation by G2DH catalysis. The yield of **14b** was not quantified because all other products were quantified using L-xylulose (**14b**) as internal standard in the quantification method. However, the stereochemical characterization of product 4-¹³C- L-xylulose (**16′b**) was done using both ¹³C NMR and chiral GC/MS (Fig 6). The peak at δ 72.0 ppm corresponds to C2 of the 2-¹³C-D-xylitol (**17′a,** Fig 6), the substrate. The reaction mixture of 2-¹³C-D-xylitol (**17′a**) with G2DH showed two additional peaks at δ 75.9 and δ 74.9 ppm in addition to the peak at δ 72.0 ppm (**Fig 6A-D**). This is a clear indication that 2-¹³C-D-xylulose (**16′a**) is not the product. Since the methoxime-TMS GC/MS method showed that it is xylulose (Fig 6A), these new peaks, not in the anomeric region, were assigned as C4 of 4-¹³C-L-xylulose (**16′b**) anomers (**16′b1, 16′b2**).

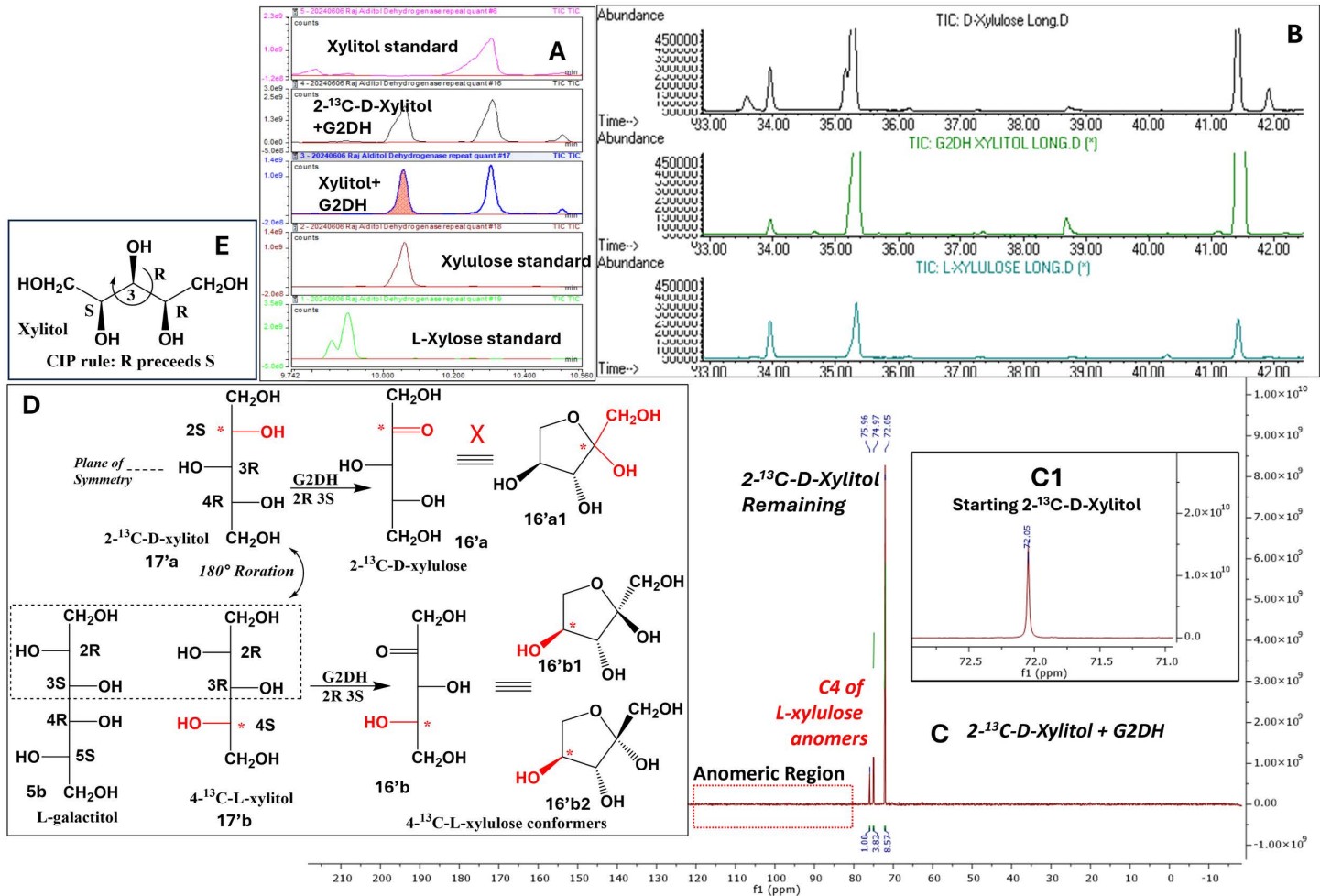

**Fig 6. Molecular characterization of product xylulose from the reaction of xylitol in the presence of galactitol-2-dehydrogenase: A:** Methoxime-TMS derivatives detected by GC/MS confirming xylulose as the product: xylitol standard (top panel), reaction mixtures panel 2 (2-¹³C-D-xylitol (**17′a**)) and 3 (xylitol (**17a,b**)), L-xylulose (**16b**) standard at panel 4 and L-xylose at panel 5 from the top. **B:** Chiral GC chromatograms showing pattern and retention times of trimethylsilyl ether (TMS) derivatives: D-xylulose (top panel, **16a**), reaction mixture (xylitol) confirming L-xylulose (**16b**) (middle panel) and L-xylulose (**16b**; bottom panel). **C:** Proton decoupled ¹³C NMR of reaction mixture (2-¹³C-D-xylitol (**16′a**) with G2DH): peak at δ 72.05 ppm corresponds to ¹³C at 2-position of **16′a (inset C1)**, peaks at δ 74.97 and δ 75.96 ppm are the C4 of α and β anomers (**16′b1**, **16′b2**) of 4-¹³C-L-xylulose (**16′b**) without any anomeric peak confirms that 2-¹³C-D-xylulose (**16′a**) is not the product. **D:** molecular mechanism showing stereochemical determinants for enantioselective L-xylulose (**16b**) formation. Notably, L-galactitol (**5b**), the substrate of the same enzyme, has the same relative conformation (D-*threo*) as 4-¹³C-L-xylitol (**16′b**) for C2 and C3 but different absolute configuration assignments. **E:** The assignment of absolute configuration of C3 in xylitol based on Cahn-Ingold-Prelog's priority rule: R-precedes S.

## Reactions with D-altritol-5-dehydrogenase

D-Altritol-5-dehydrogenase (D-A5DH or D-T2DH), which being stereospecific to 2R 3R absolute chiral systems, transformed D-altritol (**6a**) (2R 3S 4R 5R; Group III) i.e., D-talitol (**8a**) (2R 3R 4S 5R, Group III) to D-tagatose (**3a**) (0.68%) (Table 1). D-A5DH also produced D-psicose (**1a**) in this reaction in a much lower yield (0.37%). The enzyme catalyzed the oxidation of L-allitol (**10b**) (*meso*, 2R 3R 4S 5S, Group I) and D-mannitol (**11a**) (2R 3R 4R 5R, Group II), both bearing 2R 3R absolute chirality, to produce L-psicose (0.29%; **1b**) and D-fructose (0.63%; **2a**) respectively. L-talitol (**8b**) (2S 3S 4R 5S; Group III) and L-altritol (**6b**) (2S 3R 4S 5S; Group III) produced a mixture of L-tagatose (**3b**) (0.74%) and L-psicose

(0.65%; **1b**) respectively. We could detect all the products by GC/MS using the methoxime-TMS derivatization protocol. However, low product yield did not allow us to confirm the enantiomeric identity of D-/L-psicose (**1a,b**) produced from L-allitol (**10b**) and D-altritol/D-talitol (**6a, 8a**) respectively by chiral GC/MS. It is noteworthy that the stereochemical rationale suggests the enantiomeric identity presented here.

## Reaction with D-sorbitol-2-dehydrogenase

D-sorbitol-2-dehydrogenase (D-S2DH) catalyzed the oxidation of D-gulitol (**7a**) (2S 3S 4S 5R, Group IV) to D-sorbose (**4a**). It also catalyzed oxidation of L-talitol (**8b**; 2S 3S 4R 5S, Group III) to L-tagatose (**3b**). None of the other alditols produced ketose (Table 1).

This work did not optimize essential reaction conditions for each enzyme separately and the reaction conditions were only optimized for G2DH with galactitol (**5a,b**) [20–22]. However, this approach allowed us to eliminate any possible influence of variability in reaction conditions as a function of substrate ambiguity and promiscuity. The objective of this study was to investigate stereochemical determinants of the substrates towards enzyme stereospecificity, so no enzyme kinetic study was performed. However, the progress of reactions were monitored by measuring the absorbance (340 nM) of NADH (product from cofactor $NAD^+$).

## Discussion

Our work demonstrates that some, not all, alditol-2-dehydrogenases are promiscuous in the oxidation of alditols to ketoses. Two modalities of promiscuity were observed (Fig 7). The type-I promiscuity occurs when the partial stereospecificity of enzymes catalyze transformation alditols with a similar C2-C3 configuration but different C4-C5 configurations. This is clearly a pattern where stereochemical ambiguity, offered by the substrate, is used by partial stereospecificity of enzymes to reveal their promiscuity. For example, G2DH catalyzes oxidation of D-iditol (**12a**; Group II), D-altritol (**6a**; Group III) and D-gulitol (**7a**; Group IV) apart from L-galactitol (**5b**; Group I) and they all have 2R 3S absolute configuration or D-*threo* relative configuration at C2-C3. Type-I promiscuity follows Izumoring hypothesis and is also associated with higher yields. The type-II promiscuity was observed where enzyme catalyzed oxidation of alditols does not match the stereospecificity of the enzyme. Therefore, promiscuity was observed, together with substrate ambiguity. The examples are associated with Group I, Group III and IV hexitols, where L-allitol (**10b**) (2R 3R), L-talitol/altritol (**8b, 6b**) (2S 3S/2S 3R) and L-gulitol (**7b**) (2R 3R) were transformed into respective ketoses by G2DH *albeit* in lower yields (Table 1). Also, Group I and Group III alditols have opposite chirality (RS/SR) associated with C3-C4 unlike L-gulitol (**7b**) (C3-C4 has RR configuration) justifying lowest yield of L-sorbose (**4b**; 1.6%). Importantly, the type-II promiscuity is an exception to conventional understanding and the Izumoring hypothesis.

Nevertheless, other Group II alditols, with least substrate ambiguity, like D- & L-mannitol (**11a,b**) (2R 3R 4R 5R & 2S 3S 4S 5S) and L-iditol (2S 3R 4R 5S) (**12b**) neither match 2R 3S absolute chirality or D-*threo* relative configuration nor do they have opposite chirality at C3-C4 and preserve their alditol/ enantiomeric identity after180° rotation in plane, and did not undergo oxidation with promiscuous G2DH.

The oxidation of D-altritol (**6a**, Table 1) (2R 3S 4R 5R; Group III) i.e., D-talitol (**8a**) (2R 3R 4S 5R, Group III; after 180° rotation in plane) to D-tagatose (**3a**) is justified by the stereospecificity (2R 3R absolute configuration/L-*erythro* relative configuration) of D-Altritol-5-dehydrogenase (D-A5DH or D-T2DH). Remarkably, D-A5DH also produced D-psicose (**1a**) in this reaction as mixture revealed a promiscuous behavior of type-II towards 2R 3S systems. As expected, the enzyme catalyzed the oxidation of L-allitol (**10b**) (*meso*, 2R 3R 4S 5S, Group I) and D-mannitol (**11a**) (2R 3R 4R 5R, Group II), both bearing 2R 3R absolute configuration, to produce L-psicose (**1b**) and D-fructose (**2a**) respectively thereby revealing type-I promiscuity. These reactions indicate the allowed partial stereoselectivity (2R 3R) of D-A5DH. L-talitol (**8b**) (2S 3S 4R 5S; Group III) and L-altritol (**6b**) (2S 3R 4S 5S; Group III) produced a mixture of L-tagatose (**3b**) (0.74%) and L-psicose (0.65%; **1b**). Another example of type-II promiscuity of A5DH was found over enantiomers, i.e., the enzyme specific for D-altritol

 

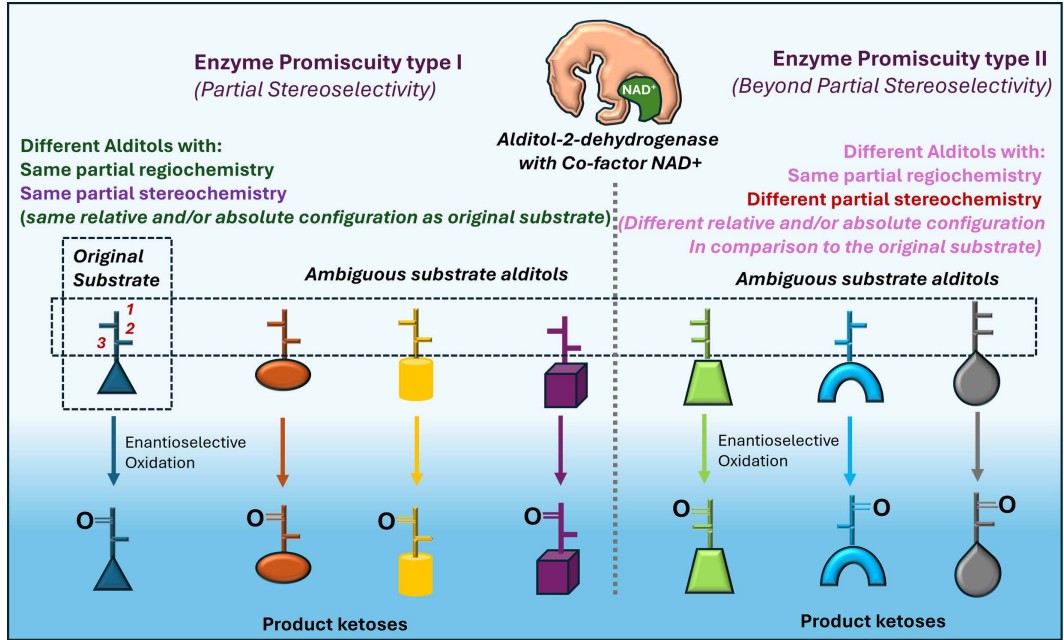

**Fig 7. Generalized concept of promiscuity types.** Alditol-2-dehydrogenase enzymes with partial regio and stereoselectivity.

(**3a**) also catalyzing oxidation of L-altritol (**14a**). These results suggest a C3-C4 bond-rotation and induced-fit-type substrate recognition mechanism may be the possible reasons for ambiguity and promiscuity. The key observation is that the enzyme and substrate, both flexible/adaptable to undergo necessary structural adjustments and show promiscuous behaviors.

D-sorbitol-2-dehydrogenase (D-S2DH) catalyzed the oxidation of D-gulitol (**7a**) (2S 3S 4S 5R, Group IV) to D-sorbose (**4a**) and L-talitol (**8b**) (2S 3S 4R 5S, Group III) to L-tagatose (**3b**), consistent with C2-C3 stereospecificity. Although D-S2DH was presumed to be stereospecific to 2S 3S/D-*erythro* absolute/relative configurations, it, however, did not produce detectable quantity of ketoses from L-mannitol (**11b**) (2S 3S 4S 5S, Group II) or D-allitol (**10a**) (*meso*, 2S 3S 4R 5R, Group I). The absorbance (S3 Fig) monitoring showed some reactivity of allitol (**10a,b**; Group I), L-iditol (2S 3R 4R 5S, Group II) (**12b**) and D-talitol/ D-altritol (**8a**, **6a**) (2R 3R 4S 5R/2R 3S 4R 5R; Group III) with D-S2DH. Possibly, the product ketoses were below detection limit.

Importantly, we could synthetically get access to the production of both enantiomers of 4x ketohexoses by these three alditol dehydrogenases and from the cohort of 10x alditols. In this experimental setup the enzymes showed different degrees of promiscuous nature. This clearly indicates that all alditol-2-dehydrogenases may not be promiscuous. While enzyme promiscuity is essential for adherence to the Izumoring hypothesis, it extends beyond the relative or absolute configurations at C2–C3. Factors such as partial stereospecificity, opposing chirality at C3–C4, bond rotation, and induced-fit mechanisms are all likely contributing elements. The observed enzyme promiscuity, manifested as slack stereospecificity or substrate adaptability, appears to stem inherently from the nature of the substrate itself. Galactitol (**5a,b**) is a *meso* hexitol and galactitol-2-dehydrogenase has shown the most promiscuous behavior in terms of substrate diversity, among the three enzymes tested. Similarly, the entire Izumoring hypothesis was built on ribitol-2/4-dehydrogenase and xylitol-2/4-dehydrogenase enzymes, involving *meso* pentitols ribitol (**13a,b**) and xylitol (**16a,b**) with a plane of symmetry in both. Therefore, *meso*-alditol-2-dehydrogenase enzymes are expected to be most promiscuous.

The example of $^{13}$C-labeled-D-xylitol (**13a;** Fig 6) (*meso*, 2R 3R 4S), the only representative pentitol investigated, producing 4-$^{13}$C-L-xylulose (**16′b**) by G2DH catalyzed C2-oxidation, may not be the result of stereochemical ambiguity.

4-$^{13}$C-L-Xylitol (**17′b**) has relative C2-C3 configuration the same as L-galactitol (D-*threo*; **5b;** Fig 6D) and has a plane of symmetry as well. But according to Cahn-Ingold-Prelog's priority rules (R precedes S) the absolute configuration of C3 had to be assigned as 3R, thereby revealing inter-conflicting nature of nomenclature systems [32].

## Conclusion

In conclusion, all enantiopure ketohexoses can be produced by the three enzymes and a cohort of ten chosen enantiopure alditols. Enzymes such as G2DH exhibited partial stereospecificity, focusing on specific configurations (C2-C3 absolute/relative) rather than the entire molecule. The enzymes D-A5DH and D-S2DH are intrinsically more stereospecific although follows the same C2-C3 configuration selectivity. Product yields are dependent on the matching of the absolute configuration of C2-C3 thereby revealing partial stereoselectivity for two carbons instead of whole molecules. Notably, substrates with variations at the C4-C5 positions exhibit lower yields compared to the original substrate. For example, in G2DH-catalyzed reactions, the observed yields are: galactitol (55%) > L-glucitol (18.5%) > D-iditol (17%) > D-altritol (11%). While all these alditols share a 2R 3S configuration, differences in their C4 and C5 stereochemistry account for the variation in enzymatic conversion. The *meso*-alditol dehydrogenases such as G2DH exhibited notable substrate flexibility, oxidizing alditols even when their C2–C3 stereochemistry differs from that of the preferred substrate. G2DH, for example, oxidizes L-allitol (2%; 2R 3R), D-talitol (3.2%; 2R,3R), L-talitol (2%; 2S 3S) and L-gulitol (1.6%; 2R,3R), illustrating catalytic promiscuity that extends beyond partial stereospecificity. A key factor seems to be the presence of a molecular plane of symmetry, present in galactitol as well as xylitol and ribitol, which likewise correlates with the broad substrate scope observed for galactitol-2-dehydrogenase, xylitol-2/4-dehydrogenase, and ribitol-2/4-dehydrogenase. Also, the promiscuous stereospecificity of the enzymes were mainly manifested when stereochemical ambiguities are present in the structure of substrate alditols. This observation is further supported by the fact that the unambiguous alditols like D-/L-mannitol (**11a,b**) or D-/L-iditol (**12a,b**) did not undergo oxidation when the enzyme stereospecificity (absolute configuration of C2-C3) did not match. As promiscuous enzymes offer potential for producing diverse ketohexoses, optimizing reaction conditions, and understanding the stereochemical interplay are crucial for their effective use in biosynthetic platforms. While mechanistic insights into SDR promiscuity and substrate ambiguity can guide the development of efficient routes for producing rare sugars, the use of R/S nomenclature over traditional D/L or *erythro*/*threo* classifications offers a more precise framework for understanding enzyme-substrate interactions. This work provides a foundation for predicting and leveraging enzyme promiscuity and substrate ambiguity in synthetic biology. Importantly, the understanding of enzymatic promiscuity and substrate ambiguity through the extent of adaptability is likely to help unravel, in future, their critical roles in evolutionary processes of enabling enzymes to acquire novel functionalities.

## Experimental procedures

### Protein expression constructs

Amino acid sequences from previous publications on galactitol-2-dehydrogenase [24–26], D-altritol-5-dehydrogenase [27] and D-sorbitol-2-dehydrogenase [28–30] (UniProt Accessions: C0KTJ6, Q89FN7, and A9CES3 respectively) were used for the generation of expression constructs. *E. coli*-optimized coding sequences were synthesized by GenScript. The nucleotide coding sequence for each protein was modified by the addition of an N-terminal His-tag, plus 5′ and 3 ′ restriction enzyme sites (GatDH and BjSDH: 5′ NdeI, 3′ HindIII; AltDH: 5′ NdeI, 3′ XhoI). Synthesized genes were cloned into the pET-30a(+) expression vector by restriction enzyme digest using the aforementioned sites (GenScript). Plasmid preparations and insert sequence verification were performed by GenScript.

### *In-vitro* cell-free enzyme expression and Ni-column purification

The NEBExpress Cell-free E. coli Protein Synthesis System (New England Biolabs) was used for in-vitro enzyme expression. Reactions were set up according to the kit protocol, with 500 ng of purified vector per 100 µL reaction. Reactions

were incubated with shaking (175 rpm) at 37 °C for 10 hours. Protein was purified using NEBExpress Ni Spin Columns (New England Biolabs) and the accompanying protocol, with elution volumes of 110 µL. Size confirmation for each protein was performed by SDS-PAGE run at 200V for 50 minutes using NuPage 12% Bis-Tris Mini-Gels (Invitrogen) and Precision Plus Protein Unstained Standards (Biorad) ladder, followed by visualization with InstantBlue Coomassie Protein Stain (Abcam) (Pictures of Gels are presented in S5 Fig). Proteins concentrations were determined using a modification of the Bradford method provided by the Pierce Bradford Plus Protein Assay Kit (Thermo Scientific), with absorbance at 595 nm measured on a TECAN Spark.

### Chemo-enzymatic synthesis of ketoses without catalytic regeneration

In an Eppendorf tube (1.5 mL), Alditol (TCI and Omicron Bio, usually $C_xH_{2x+2}O_x$, 0.1 M in water, 22 µL; ~400 µg) was added with HPLC grade water (21 µL), sodium carbonate (Mallinckrodt)-bicarbonate (Fisher Scientific) buffer (1 M, pH~10, 20 µL), Magnesium chloride (Fisher Scientific, $MgCl_2$, 0.002 M in water, 5 µL), Nicotinamide Adenine Dinucleotide (Roche, $NAD^+$, 0.1 M, 22 µL), alditol dehydrogenase (generated cell-free, 1–2 µg in 10 µL). The reaction mixture was incubated at 30 °C for 24 h.

The reaction was initially optimized using the 96-well plate format. The progress of reaction was monitored by measuring the absorbance (340 nM) of NADH using a TECAN Spark at a controlled temperature of 30 °C until the period at which no further increment in NADH was observed (S6 Fig).

### Chemo-enzymatic synthesis of ketoses with catalytic regeneration

In an Eppendorf tube (1.5 mL), Alditol (TCI and Omicron Bio, usually $C_xH_{2x+2}O_x$, 0.1 M in water, 22 µL; ~400 µg) was added with HPLC grade water (20 µL), sodium carbonate(Mallinckrodt)-bicarbonate (Fisher Scientific) buffer (1 M, pH~8.5, 10 µL), Magnesium chloride (Fisher Scientific, $MgCl_2$, 0.002 M in water, 5 µL), Nicotinamide Adenine Dinucleotide (Roche, $NAD^+$, 0.1 M, 8 µL), Alditol dehydrogenase (generated cell-free, 1–2 µg in 10 µL), sodium pyruvate (Sigma-Aldrich, 0.81 M in water, 3 µL) and lactate dehydrogenase (Sigma-Aldrich, 0.4 µg in 10 µL). The reaction mixture was incubated at 30ºC for 24 h.

### Quantitative detection of product ketoses by GC/MS

The Reaction mix (100 µL) was diluted with cold 90% Aqueous-Ethanol (Water: EtOH = 1:9; 900 µL). Stored at −20ºC for 4 h after thorough mixing. Centrifuged (10000 rpm) for 10 min. An aliquot of 20 µL (except 3 µL for G2DH +Galactitol (**5a,b**)) was taken in GC-vial (2 mL) and dried under $N_2$-stream. The dried sample was first resuspended in methoxyl amine hydrochloride (Sigma-Aldrich) in pyridine (Sigma-Aldrich) solution (50 µL, 25 mg/mL), incubated at 60°C for 45 min, sonicated for 10 min, and incubated for an additional 45 min at 60°C. Second, 50 µL of N-methyl-N-trimethylsilyl trifluoroacetamide with 1% trimethylchlorosilane (MSTFA+1% TMCS, Thermo Scientific) was added and samples were incubated at 60 °C for 35 min, cooled to room temperature, and was transferred to a 150-µL glass insert and placed back in the corresponding GC-MS autosampler vial.

### GC/MS calibration curve for ketoses

Two Ketose standard mix solutions (0.5 mL, 100 µg/mL) were prepared by mixing D-fructose (**2a**) (50 µL; 1 mg/mL) with D-tagatose (**3a**; 50 µL; 1 mg/mL) and separately, D-psicose (**1a**) (50 µL; 1 mg/mL) with D-sorbose (**4a**; 50 µL; 1 mg/mL) and diluting them with miliQ water (400 µL). The standard mixtures were further 2xfold serially diluted (250 µL standard + 250 µL miliQ water) to 50, 25, 12.5, 6.25, 3.125, 1.5625, 0.78125 µg/mL. Aliquotes (20 µL; i.e., 2.0, 1.0, 0.5, 0.25. 0.125, 0.0625, 0.03125, 0.015625 µg in vial) from each dilution was taken in GC-vial and L-Xylulose (**16b**, Fig 6) in water (0.3 µg/mL; 6 µL) was added to each vial as the internal standard. The vials were placed under $N_2$-stream, dried and derivatized as mentioned above. The peak ratio of each ketose with internal standard were plotted to generate calibration

curves. The R [2] for every fatty acid component can be found on the plots. The limit of detection was calculated with the respective regression analyses. Details can be found in the (S2 Fig). Detailed calculations are provided in S3 Dataset.

## GC/MS method for monosaccharide-oxime-TMS derivative detection

Samples (1 µL) were injected (into a TSQ8000 Evo GCMS instrument) at a 20:1 split ratio to a 30 m DB-5MS column (Agilent, 0.25 mm i.d., 0.25 µm film thickness) with a 1.2 mL/min helium gas flow rate. GC inlet was held at 285°C. The oven program started at 80°C for 30 s, followed by a ramp of 15 °C/min to 330 °C, and a 7 min hold. Masses between 50–650 m/z were scanned at 5 scans/sec under electron impact ionization. Transfer line and ion source were held at 300 and 260 °C, respectively. All chromatograms have been presented in S1 Fig.

## Qualitative detection of ketose enantiomers by GC/MS

Samples (20 µL), as processed for quantitative assays mentioned earlier, were taken in GC-vials (2 mL) and dried under $N_2$-stream. The dried sample was first resuspended in pyridine (Sigma-Aldrich, 50 µL)), incubated at 60°C for 25 min, sonicated for 10 min, and incubated for an additional 15 min at 60°C. Second, 50 µL of N-methyl-N-trimethylsilyl trifluoro-acetamide with 1% trimethylchlorosilane (MSTFA + 1% TMCS, Thermo Scientific) was added and samples were incubated at 60 °C for 35 min, cooled to room temperature, and was transferred to a 150-µL glass insert and placed back in the corresponding GC-MS autosampler vial. Chromatogram plots are presented in S4 Fig.

GC/MS method for ketose-enantiomer-TMS derivative detection using chiral column.

Samples (4 µL) were injected (by Agilent 7683 series injector into Agilent 6890 GC system with Agilent 5973 Network Mass selective detector GC/MS instrument) at a splitless mode ratio to a 25 m CP-Chirasil-Dex CB column (Agilent CP7502, 0.25 mm i.d., 0.25 µm film thickness) with a 5 mL/min helium gas flow rate. GC inlet was held at 200 °C. The oven program started at 40 °C for 1.0 min, followed by a ramp of 5 °C/min to 140 °C, and a 1.0 min hold. A further ramp with 2 °C/min to 185 °C and a hold for 1.0 min. Masses between 50–700 m/z were scanned with a solvent delay of 10.0 min) under electron impact ionization. Transfer line and ion source were held at 225 °C and 230 °C, respectively.

$^{13}$C-NMR for product stereochemistry detection for chosen 2-$^{13}$C-labeled alditols.

The reaction mixture (in water/buffer 100 µL) and standard 2-$^{13}$C-labeled alditols (starting material, 400 µg) were taken up to 500 µL sterile water and $D_2O$ (50 µL) was added to it. The 13CNMR was recorded on Bruker Neo400 instrument (400MHz) with a Prodigy BBFO probe using in-built$^{13}$C CPD (proton decoupled) in $H_2O$ + 10% $D_2O$ program (100.61 MHz) with 256 scans. The spectra were processed in MestreNova LITE CDE software. The product from oxidation of 2-$^{13}$C-D-galactitol (**5′a**) by G2DH, i.e., 5-$^{13}$C-L-tagatose (**3′b**) was purified and $^{13}$C NMR was recorded. The NMR spectra and purification protocol were presented as S6 Fig.

## NMR characterization data

**2-$^{13}$C-L-glucitol** (0.4 mg in 0.55 mL solvent):

13 C-CPD NMR (10% $D_2O$ in water, 100 MHz) δ ppm: 73.0 ($^{13}$C2)

**2-$^{13}$C-L-glucitol reaction mixture G2DH with 2-$^{13}$C-L-fructose product** (0.4 mg in 0.55 mL solvent):

13 C-CPD NMR (10% $D_2O$ in water, 100 MHz) δ ppm: 73.0 ($^{13}$C2, unreacted 2-$^{13}$C-L-glucitol), 98.1 ($^{13}$C2, β-L-fru*p*, 2-$^{13}$C-L-fructose, product anomer), 101.5 ($^{13}$C2, β-L-fru*f*, 2-$^{13}$C-L-fructose, product anomer), 104.4 ($^{13}$C2, α-L-fru*f*, 2-$^{13}$C-L-fructose, product anomer)

**2-$^{13}$C-D-xylitol** (**17′a**)(0.4 mg in 0.55 mL solvent):

13 C-CPD NMR (10% $D_2O$ in water, 100 MHz) δ ppm: 72.0 ($^{13}$C2)

**2-$^{13}$C-D-xylitol** (**17′a**) **reaction mixture (G2DH) with 4-$^{13}$C-L-xylulose (16′b) product** (0.4 mg in 0.55 mL solvent):

13 C-CPD NMR (10% $D_2O$ in water, 100 MHz) δ ppm: 72.0 ($^{13}$C2, unreacted 2-$^{13}$C-D-xylitol), 74.9 ($^{13}$C4, α or β-L-xyl*f*, 4-$^{13}$C-L-xylulose, product anomer), 75.9 ($^{13}$C4, α or β-L-xyl*f*, 4-$^{13}$C-L-xylulose, product anomer)

**2-¹³C-D-galactitol (5′a)** (0.4 mg in 0.55 mL solvent):

13 C-CPD NMR (10% D$_2$**(5′a)**O in water, 100 MHz) δ ppm: 70.2 (¹³C2)

**2-¹³C-D-galactitol (5′a) reaction mixture (G2DH) with 5-¹³C-L-tagatose product (3′b)** (0.4 mg in 0.55 mL solvent):

13 C-CPD NMR (10% D$_2$O in water, 100 MHz) δ ppm: 66.4 (¹³C5, β-L-tag*f*, 5-¹³C-L-tagatose, product anomer), 69.3 (¹³C5, α-L-tag*f*, 5-¹³C-L-tagatose, product anomer), 70.2 (¹³C2, unreacted 2-¹³C-D-galactitol)

## Supporting information

**S1 Fig. Methoxime-TMS Derivatization scheme for GC/MS detection of ketoses; tautomers in equilibrium (L-fructose (2b) is an example); Chromatograms of reaction mixture of alditols with three enzymes.**
(DOCX)

**S2 Fig. GC/MS chromatogram explaining peaks assigned for quantitation of ketoses.** The linear plot of Ketose/xylulose (internal standard) GC/MS intensity ratio *vs* concentration. LOD was calculated from regression analysis.
(DOCX)

**S3 Dataset. Detailed calculations of Ketose products quantitation.**
(XLSX)

**S4 Fig. GC/MS Chromatograms of standard Ketose enantiomers derivatized with trimethylsilyl groups.** GC/MS characterization of enantiopure ketose products from alditols with **G2DH/D-A5DH/D-S2DH** reactions.
(DOCX)

**S5 Fig. Gel electrophoresis of purified enzymes.**
(DOCX)

**S6 Fig. The initial reaction progress was monitored by measuring the absorbance (340 nM) of NADH using a TECAN Spark.**
(DOCX)

**S7 Fig. Purification protocol of 5-¹³C-L-tagatose and Proton decoupled 13C NMR spectra of 2-¹³C-D-galactitol (top), Product 5-¹³C-L-tagatose (middle) and unlabeled L-tagatose (bottom panel).**
(DOCX)

## Acknowledgments

The authors thank the Analytical Resources Core (RRID: SCR_021758) at Colorado State University for instrument access. Thanks to NREL, CSU laboratory.

## Author contributions

**Conceptualization:** Prithwiraj De, Jenna Salvat, Eliza Walthers, Michael Wells, Claudia M. Boot.

**Formal analysis:** Prithwiraj De, Jenna Salvat, Michael Wells.

**Funding acquisition:** Richard T. Conant, Claudia M. Boot.

**Investigation:** Prithwiraj De, Jenna Salvat, James Henriksen, Michael Wells, Claudia M. Boot.

**Methodology:** Prithwiraj De, Eliza Walthers, James Henriksen, Michael Wells, Richard T. Conant, Claudia M. Boot.

Project administration: Richard T. Conant.

**Resources:** Prithwiraj De, Michael Wells, Claudia M. Boot.

**Supervision:** Prithwiraj De, James Henriksen.

**Validation:** Prithwiraj De.

**Visualization:** Prithwiraj De.

**Writing – original draft:** Prithwiraj De, Michael Wells.

**Writing – review & editing:** Prithwiraj De, Jenna Salvat, Eliza Walthers, James Henriksen, Michael Wells, Richard T. Conant, Claudia M. Boot.

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
