## [Decision Letter · Decision Letter 0]

PONE-D-25-13867Harnessing Enzyme Promiscuity of Alditol-2-Dehydrogenases for Oxidation of Alditols to Enantiopure KetosesPLOS ONE

Dear Dr. De,

Thank you for submitting your manuscript to PLOS ONE. After careful consideration, we feel that it has merit but does not fully meet PLOS ONE’s publication criteria as it currently stands. Therefore, we invite you to submit a revised version of the manuscript that addresses the points raised during the review process.

We look forward to receiving your revised manuscript.

Kind regards,

Vinod Kumar Vashistha

Academic Editor

PLOS ONE

Journal Requirements:

2. We note that this submission includes NMR spectroscopy data. We would recommend that you include the following information in your methods section or as Supporting Information files:

1) The make/source of the NMR instrument used in your study, as well as the magnetic field strength. For each individual experiment, please also list: the nucleus being measured; the sample concentration; the solvent in which the sample is dissolved and if solvent signal suppression was used; the reference standard and the temperature.

2) A list of the chemical shifts for all compounds characterised by NMR spectroscopy, specifying, where relevant: the chemical shift (δ), the multiplicity and the coupling constants (in Hz), for the appropriate nuclei used for assignment.

3)The full integrated NMR spectrum, clearly labelled with the compound name and chemical structure.

We also strongly encourage authors to provide primary NMR data files, in particular for new compounds which have not been characterised in the existing literature. Authors should provide the acquisition data, FID files and processing parameters for each experiment, clearly labelled with the compound name and identifier, as well as a structure file for each provided dataset. See our list of recommended repositories here: https://journals.plos.org/plosone/s/recommended-repositories

Funding for this research was supported by the Keck Research Foundation, the Grantham Foundation, and Grant #2000-67030-31475 from the US Department of Agriculture

Reviewers' comments:

Reviewer's Responses to Questions

**Comments to the Author**

1. Is the manuscript technically sound, and do the data support the conclusions?

Reviewer #1: Partly

Reviewer #2: Partly

2. Has the statistical analysis been performed appropriately and rigorously? 

Reviewer #1: No

Reviewer #2: N/A

3. Have the authors made all data underlying the findings in their manuscript fully available?

Reviewer #1: Yes

Reviewer #2: Yes

4. Is the manuscript presented in an intelligible fashion and written in standard English?

Reviewer #1: No

Reviewer #2: Yes

5. Review Comments to the Author

Reviewer #1: The manuscript presents a promising investigation into enzyme stereospecificity and its application in synthesizing enantiopure ketohexoses from alditols. While the core concept and experimental direction are scientifically relevant, the overall presentation requires significant refinement. Below are my observations on the manuscript:

1. The abstract should begin with a clear and concise statement of the problem, follow with the methodology and main results, and end with the implications and novelty. Reducing jargon and tightening the structure would make it more accessible and impactful. Some observations on the abstract:

i. The abstract is dense and difficult to follow in places due to convoluted sentence structure. The main objectives and findings are buried under complex phrasing. Consider simplifying and clarifying key statements for better readability.

ii. The phrase "this work is not kinetic study but an unconventional stereochemical investigation..." is awkwardly placed at the end and sounds like an afterthought. A clearer and more formal way to state this distinction would be beneficial.

iii. While the abstract emphasizes enzyme promiscuity and stereochemical insights, it lacks specific examples or quantitative findings to support the claims. Including a concise result or two (e.g., yields or selectivity ratios) would enhance its impact.

iv. The structure of the sentence beginning with “For instance, G2DH oxidizes not only galactitol…” is overly long and confusing. Breaking it into two sentences would improve clarity.

v. A reader unfamiliar with the enzymes or the broader context of the study may struggle to understand the significance of the findings. Briefly stating the broader application (e.g., in rare sugar synthesis or biotechnology) would help ground the study's relevance.

2. The Introduction is quite long and dense, covering a wide range of topics from sugar stereochemistry to enzyme promiscuity. It would benefit from better organization and condensation to maintain focus and improve readability.

3. While the Introduction provides a comprehensive background, it does not clearly articulate the specific research gap that this study addresses. The authors should clearly state what is unknown or unresolved in the field and how their work fills that gap.

4. The novelty of the present study is mentioned at the end but not emphasized enough throughout the introduction. It should be clearer how this study advances the current understanding of enzyme promiscuity or rare sugar synthesis.

5. Several concepts (e.g., enzyme promiscuity, stereochemistry at C2/C3) are repeated multiple times in slightly different wording. These points could be consolidated to enhance clarity and avoid redundancy.

6. The transition between topics—such as moving from monosaccharide classification to enzyme mechanisms—is abrupt in places. The authors should consider using linking sentences to improve logical flow between paragraphs.

7. Although many references are cited, some are not well integrated into the narrative. Rather than simply listing studies, the authors should briefly describe what each reference contributes to support their claims.

8. Some sentences are long and convoluted, which makes comprehension difficult. For example:

"We wanted to utilize promiscuity to synthetically access diverse products but realized that this can also create challenges..."

This could be rewritten more concisely and precisely.

9. The results section primarily provides descriptive interpretations of enzyme activity and stereochemistry but lacks quantitative data (e.g., reaction yields, specific activity, enzyme kinetics such as Km and Vmax). Including these values is crucial for evaluating the efficiency and specificity of the enzymes studied.

10. The claim that these enzymes exhibit promiscuous activity is intriguing but unsubstantiated by experimental data. The manuscript should present experimental evidence (e.g., conversion rates for non-canonical substrates) to support the assertion of enzymatic promiscuity.

11. Several interpretations are based on planar rotation and structural symmetry (e.g., 180° rotation of alditols), which are theoretically sound but require experimental validation. The manuscript would benefit from structural confirmation, such as NMR or X-ray data, to substantiate these rotational assumptions.

12. While the text references “Fig 2” and “Table 1,” it lacks clear structural illustrations or schemes to support the stereochemical discussion. Stereochemical relationships and configurations are better communicated with diagrams—without them, the textual analysis is hard to follow.

13. The results use terms like “L-threo,” “D-erythro,” and absolute configurations (e.g., 2R 3S) interchangeably, which can be confusing without consistent definitions or clear mapping.

14. The suggestion to rename D-A5DH as “D-talitol-2-dehydrogenase” introduces ambiguity. The re-naming should only be proposed if supported by clear comparative activity data and substrate specificity assays that justify this new identity.

15. While the enzymes are said to have “never been investigated for their promiscuous role,” it is unclear what new findings this study offers beyond theoretical considerations.

16. The section occasionally repeats the same structural rotation logic across multiple examples without introducing new insights. This results in redundancy that could be streamlined for better clarity and impact.

17. The application of the CIP rules is a good start, but the results could be strengthened by comparing these assignments to known experimental stereochemical data (e.g., crystallography or chiroptical properties). Currently, the application feels superficial and disconnected from experimental support.

18. The mention that pentitols and tetritols deviate from the observed configuration rules is an important observation but left underexplored. The authors should expand on this with examples or explanations to avoid leaving critical stereochemical exceptions unresolved.

19. The authors should consider reorganizing the results around individual enzymes or alditol groups to facilitate readability.

20. he biological or mechanistic significance of this classification remains somewhat abstract. A clearer link between structure group and enzyme specificity would strengthen the impact.

21. The yields of ketose products are reported without any mention of replicates, error bars, or statistical analysis. Are these results reproducible? Were the yields consistent across replicates? Quantitative results must be supported with standard deviations or confidence intervals.

22. For D-A5DH and D-S2DH, the product yields are extremely low (<1%), and there is no discussion on the biological relevance or catalytic efficiency. Since optimization was not performed for these enzymes, any comparison with G2DH is weak. Either justify the inclusion of these poorly performing systems or elaborate more thoroughly on their significance despite low yield.

23. There are multiple mentions of overlapping peaks and the need for chiral columns or unique derivatization to resolve product identities. However, the section lacks validation of the separation method (e.g., resolution factors, retention times). It is unclear whether the minor peaks are confidently identified.

24. Many substrates yield very low percentages of ketose products (1–2%), yet the manuscript does not critically analyze why. Is it due to poor substrate binding, stereochemical hindrance, or enzyme inefficiency? Such discussion is essential for a deeper understanding of enzyme-substrate interactions.

25. he authors state that enzyme kinetics were not performed, yet they attempt to draw conclusions on enzyme specificity. Without kinetic data (e.g., Km, Vmax), it is difficult to meaningfully discuss substrate preference or stereospecificity.

26. The authors suggest stereospecific outcomes based on product configuration, but in cases where enantiomeric identity was not confirmed (e.g., low-yield products not resolved by chiral GC/MS), these claims remain speculative. The language must reflect this uncertainty.

27. The use of NADH absorbance at 340 nm to monitor reaction progress is mentioned briefly. More details are needed: How was baseline drift handled? Were there interfering signals? Was a standard curve used for quantification?

28. Important results (e.g., product yields, structural relationships) are reported without sufficient integration with Table 2 or corresponding figures. Key findings must be directly tied to data tables/visuals within the text.

29. The explanation of stereochemical preferences and exceptions needs to be more coherent and visually supported by clear diagrams or tables. Without this, the logic behind "partial stereospecificity" remains vague.

30. A concise introduction or a schematic summarizing the hypothesis would help contextualize the findings, especially for readers less familiar with carbohydrate enzymology.

31. While the authors suggest that bond rotation, induced fit, and substrate flexibility contribute to promiscuity, these are presented speculatively without experimental or computational support. A deeper mechanistic exploration or references to literature supporting these claims would strengthen the conclusions.

32. Several statements are repetitive, particularly in describing C2-C3 and C3-C4 configurations. The discussion could benefit from tighter editing to reduce redundancy and improve readability.

33. The finding that some enzymes are promiscuous while others are not raises interesting questions about enzyme evolution, specificity, and substrate scope, which are not addressed. Discussing whether these promiscuous behaviors have physiological relevance or evolutionary drivers would elevate the impact of the findings.

34. The promiscuity of galactitol-2-dehydrogenase and the reference to meso substrates as being inherently more prone to promiscuous oxidation is an interesting observation but lacks structural justification. What about meso-structure facilitates promiscuity? A deeper stereoelectronic explanation would help here.

35. The discussion around 13C-labeled xylitol is confusing and does not clearly support or challenge earlier conclusions. The commentary on stereochemical nomenclature (R vs. S) is convoluted and would be better placed in a footnote or supplementary section.

36. Several points are reiterated throughout the conclusion, such as the enzyme stereospecificity toward C2-C3 configurations and the adaptability of meso-alditol dehydrogenases. These could be consolidated to enhance clarity and avoid redundancy.

37. The statement about unraveling "critical roles in evolutionary processes" seems speculative and not directly supported by the presented data. The authors should either provide stronger evidence for such a claim or reframe it as a hypothesis for future investigation.

38. Phrases like "the quality of adaptability" and "loose stereospecificity" are vague. It would be helpful if the authors clarified what is meant by "quality" in this context and provided more precise biochemical definitions or metrics where applicable.

39. The conclusion could be strengthened by briefly summarizing key numerical or comparative results (e.g., yields, enzyme efficiencies, conversion rates), rather than relying entirely on qualitative descriptors.

To enhance the manuscript's quality and scientific rigor, the authors should revise the conclusion for precision, structure, and coherence—emphasizing key results, clarifying terminology, and maintaining alignment with the data. Moreover, a clearer articulation of novelty, stronger linkage between experimental outcomes and broader implications, and acknowledgment of limitations is needed. A major revision is recommended to improve clarity, analytical depth, and the overall effectiveness of communication.

Reviewer #2: The authors have explored the promiscuous nature of alditol-2-dehydrogenase enzymes and clearly showed production of several enantiopure ketohexoses from enantiopure alditols. I just have a couple of questions regarding protein expression and NAD reduction assays.

Fig S4 shows the SDS-PAGE analysis of purified enzymes. I am not able to see any clear bands in the elution fractions for S2DH enzyme. Does the star symbol next to the band in FT represent the correct size for the protein? Does that mean the protein didn't bind to the resin? It would be great if the authors could clarify this. I am wondering if this is the reason why the authors didn't detect ketoses from this enzyme.

Fig S5 shows NAD+ reduction to NADH. The authors did mention that it is not a kinetic study but they can definitely calculate the specific activities of NAD+ reduction with several substrates for all three enzymes.

This is a good work and it should be published after these questions are properly addressed by the authors.

6. PLOS authors have the option to publish the peer review history of their article (what does this mean? ). If published, this will include your full peer review and any attached files.

**Do you want your identity to be public for this peer review?** For information about this choice, including consent withdrawal, please see our Privacy Policy .

Reviewer #1: No

Reviewer #2: No

---

## [Author Response · Author response to Decision Letter 1]

15 May 2025

Dr. Prithwiraj De

Postdoctoral Research Scientist

Colorado State University, Fort Collins, CO 80523

prithwiraj.de@colostate.edu; prithwiraj.de@gmail.com; phone: +1 (970) 294 6145

Re: Rebuttal letter for publication (PONE-D-25-13867)

Title: Harnessing Enzyme Promiscuity of Alditol-2-Dehydrogenases for Oxidation of Alditols to Enantiopure Ketoses

To,

Dr. Vashistha,

The Academic Editor, PLoS One

On behalf of all the authors, I, take the opportunity to sincerely thank the Academic Editor and the reviewers for their in-depth, suggestive and expansive reviews. We have now corrected, modified and ameliorated the manuscript and the narration in line with the reviewer’s suggestions. We also have modified according to journal policy. We have submitted the NMR files to BMRB database (ID 53168). Please also find our response to the reviewers with this letter.

We think that the findings and perspective of this work will help scientists push their chemical/biochemical inquiries forward and resources that will be of use to the research community when suitably published in PLoS One.

Thanking you for your kind consideration

Sincerely

Prithwiraj De

May 15, 2025

Journal Requirements:

Ans: Complied with the journal style.

2. We note that this submission includes NMR spectroscopy data. We would recommend that you include the following information in your methods section or as Supporting Information files:

1) The make/source of the NMR instrument used in your study, as well as the magnetic field strength. For each individual experiment, please also list: the nucleus being measured; the sample concentration; the solvent in which the sample is dissolved and if solvent signal suppression was used; the reference standard and the temperature.

Ans: The NMR data and instrument specifications as well as solvent and chemical shifts have been added with the method section; as Characterization data.

2) A list of the chemical shifts for all compounds characterised by NMR spectroscopy, specifying, where relevant: the chemical shift (δ), the multiplicity and the coupling constants (in Hz), for the appropriate nuclei used for assignment.

Ans: The NMR data and instrument specifications as well as solvent and other relevant data has been added with the method section.

3)The full integrated NMR spectrum, clearly labelled with the compound name and chemical structure.

Ans: The 13CNMR did not require integration.

We also strongly encourage authors to provide primary NMR data files, in particular for new compounds which have not been characterised in the existing literature. Authors should provide the acquisition data, FID files and processing parameters for each experiment, clearly labelled with the compound name and identifier, as well as a structure file for each provided dataset. See our list of recommended repositories here: https://journals.plos.org/plosone/s/recommended-repositories

Ans: the 13C NMR files have been deposited in the BMRB NMR database. BMRB ID 53168.

Funding for this research was supported by the Keck Research Foundation, the Grantham Foundation, and Grant #2000-67030-31475 from the US Department of Agriculture

Ans: Complied. The statement has been added.

Ans: Complied

Ans: Complied. The figure legends have been added.

Reviewers' comments:

Reviewer's Responses to Questions

Comments to the Author

1. Is the manuscript technically sound, and do the data support the conclusions?

Reviewer #1: Partly

Reviewer #2: Partly

2. Has the statistical analysis been performed appropriately and rigorously?

Reviewer #1: No

Reviewer #2: N/A

3. Have the authors made all data underlying the findings in their manuscript fully available?

Reviewer #1: Yes

Reviewer #2: Yes

4. Is the manuscript presented in an intelligible fashion and written in standard English?

Reviewer #1: No

Reviewer #2: Yes

5. Review Comments to the Author

Reviewer #1: The manuscript presents a promising investigation into enzyme stereospecificity and its application in synthesizing enantiopure ketohexoses from alditols. While the core concept and experimental direction are scientifically relevant, the overall presentation requires significant refinement.

Thank you for your in-depth and suggestive review. We really appreciate it.

Below are my observations on the manuscript:

1. The abstract should begin with a clear and concise statement of the problem, follow with the methodology and main results, and end with the implications and novelty. Reducing jargon and tightening the structure would make it more accessible and impactful. Some observations on the abstract:

i. The abstract is dense and difficult to follow in places due to convoluted sentence structure. The main objectives and findings are buried under complex phrasing. Consider simplifying and clarifying key statements for better readability.

ii. The phrase "this work is not kinetic study but an unconventional stereochemical investigation..." is awkwardly placed at the end and sounds like an afterthought. A clearer and more formal way to state this distinction would be beneficial.

iii. While the abstract emphasizes enzyme promiscuity and stereochemical insights, it lacks specific examples or quantitative findings to support the claims. Including a concise result or two (e.g., yields or selectivity ratios) would enhance its impact.

iv. The structure of the sentence beginning with “For instance, G2DH oxidizes not only galactitol…” is overly long and confusing. Breaking it into two sentences would improve clarity.

v. A reader unfamiliar with the enzymes or the broader context of the study may struggle to understand the significance of the findings. Briefly stating the broader application (e.g., in rare sugar synthesis or biotechnology) would help ground the study's relevance.

Ans (1i-v): Thank you for the comments. We have now modified the ‘Abstract’ aligning with these comments. Also, added a graphical abstract for the clarity.

2. The Introduction is quite long and dense, covering a wide range of topics from sugar stereochemistry to enzyme promiscuity. It would benefit from better organization and condensation to maintain focus and improve readability.

3. While the Introduction provides a comprehensive background, it does not clearly articulate the specific research gap that this study addresses. The authors should clearly state what is unknown or unresolved in the field and how their work fills that gap.

Ans (2,3): We agree. The complexity of the topic depending on several background considerations are integral parts of the work. We put our best effort into improving.

4. The novelty of the present study is mentioned at the end but not emphasized enough throughout the introduction. It should be clearer how this study advances the current understanding of enzyme promiscuity or rare sugar synthesis.

Ans: We agree. We have modified.

5. Several concepts (e.g., enzyme promiscuity, stereochemistry at C2/C3) are repeated multiple times in slightly different wording. These points could be consolidated to enhance clarity and avoid redundancy.

Ans: We agree. However, given the complexity of the stereochemical topic, we felt it was important to repeatedly remind readers of the underlying conceptual framework; especially for those less familiar with the subject. We have revised the text as much as practicable to support this aim.

6. The transition between topics—such as moving from monosaccharide classification to enzyme mechanisms—is abrupt in places. The authors should consider using linking sentences to improve logical flow between paragraphs.

Ans: We have now modified the narrative.

7. Although many references are cited, some are not well integrated into the narrative. Rather than simply listing studies, the authors should briefly describe what each reference contributes to support their claims.

Ans: We have now modified the narrative.

8. Some sentences are long and convoluted, which makes comprehension difficult. For example:

"We wanted to utilize promiscuity to synthetically access diverse products but realized that this can also create challenges..." This could be rewritten more concisely and precisely.

Ans: We have now modified the narrative.

9. The results section primarily provides descriptive interpretations of enzyme activity and stereochemistry but lacks quantitative data (e.g., reaction yields, specific activity, enzyme kinetics such as Km and Vmax). Including these values is crucial for evaluating the efficiency and specificity of the enzymes studied.

Ans: The quantitative comparison and data (reaction yield, enantiomeric identity) has been presented at the later part of the result. Since this is not a kinetic study, no specific activity/Km/Vmax was done and clearly mentioned in the manuscript. Please also see the answer to comment 25.

10. The claim that these enzymes exhibit promiscuous activity is intriguing but unsubstantiated by experimental data. The manuscript should present experimental evidence (e.g., conversion rates for non-canonical substrates) to support the assertion of enzymatic promiscuity.

Ans: The present work only experimentally assesses the promiscuity of these enzymes by their stereochemical descriptors. Otherwise, all alditol-2-dehydrogenase enzymes are considered somewhat promiscuous. As claimed by the References 20 & 21.

11. Several interpretations are based on planar rotation and structural symmetry (e.g., 180° rotation of alditols), which are theoretically sound but require experimental validation. The manuscript would benefit from structural confirmation, such as NMR or X-ray data, to substantiate these rotational assumptions.

Ans: Rotation (180º) in plane is intrinsic molecular descriptor in terms of stereochemistry. The structural characterizations with the appropriate use of 13C-labelled alditols and 13C-proton-decoupled NMR spectra along with GC/MS analyses characterized the product ketoses and enantiopurity conclusively. Two examples have been presented in the manuscript; 13C-2-L-glucitol (Fig 4) and 13C-2-D-L-xylitol (Fig 5) oxidations with G2DH; showing approach, stereospecificity and mechanism of G2DH conclusively. X-ray is beyond the scope of this work.

12. While the text references “Fig 2” and “Table 1,” it lacks clear structural illustrations or schemes to support the stereochemical discussion. Stereochemical relationships and configurations are better communicated with diagrams—without them, the textual analysis is hard to follow.

Ans: Table 1 has been modified with the concept of (column wise) common absolute/relative configurations proposed for single alditol-2-dehydrogenase oxidation.

13. The results use terms like “L-threo,” “D-erythro,” and absolute configurations (e.g., 2R 3S) interchangeably, which can be confusing without consistent definitions or clear mapping.

Ans: We agree. L-threo,” “D-erythro,” and absolute configurations (e.g., 2R 3S) had to be used repeatedly because they are equivalent to each-other only to a certain extend (i.e, for hexitols). They are not configuration wise same for pentitols. It is a complex concept and could not be presented in any different way.

14. The suggestion to rename D-A5DH as “D-talitol-2-dehydrogenase” introduces ambiguity. The re-naming should only be proposed if supported by clear comparative activity data and substrate specificity assays that justify this new identity.

Ans: D-altritol and D-talitol are chemically the same molecule and same substrate for the enzyme. It is the regioselectivity of the D-altritol-5-dehydrogenase i.e., oxidizing 5-position of D-altritol which is essentially the 2-position of D-talitol (a 180º relationship) that determine the product. We have shown that D-talitol produced D-tagatose and D-psicose as the oxidation products when reacted with D-altritol-5-dehydrogenase (D-A5DH). Therefore, we feel that calling D-A5DH as D-talitol-2-dehydrogenase was a justified description; particularly in the context of open chain configuration of alditols, partial regioselectivity and clearly demonstrates the substrate ambiguity. Although, it is not a claim.

15. While the enzymes are said to have “never been investigated for their promiscuous role,” it is unclear what new findings this study offers beyond theoretical considerations.

Ans: The first of our finding is that enzyme promiscuity is associated with different levels of substrate adaptability. The promiscuity of G2DH is not like A5DH or D-S2DH. Second, If anyone wants to choose an enzyme expecting suitability for multiple alditols/substrates, the stereochemical qualification becomes the plane of symmetry in the substrate. This information, therefore, bridges knowledge-gap for predictive synthesis of rare ketoses using enzyme promiscuity and utilize in the biotech industry.

16. The section occasionally repeats the same structural rotation logic across multiple examples without introducing new insights. This results in redundancy that could be streamlined for better clarity and impact.

Ans: Thanks for the suggestion. The rotation in plane and Fisher projection of the molecule is very crucial and the guiding tool/foundation for this investigation in terms of enantioselective product characterization. The entire cohort of alditols were practically based upon this stereochemical principle. We wanted to emphasize it enough to justify our observations in every context of this complex work.

17. The application of the CIP rules is a good start, b

---

## [Editor Report · Decision Letter 1]

Harnessing Enzyme Promiscuity of Alditol-2-Dehydrogenases for Oxidation of Alditols to Enantiopure Ketoses

PONE-D-25-13867R1

Dear Dr. De,

We’re pleased to inform you that your manuscript has been judged scientifically suitable for publication and will be formally accepted for publication once it meets all outstanding technical requirements.

Kind regards,

Vinod Kumar Vashistha

Academic Editor

PLOS ONE

Additional Editor Comments (optional):

Authors have significantly reviewed the manuscript as per reviewer's comments.
---

## [Editor Report · Acceptance letter]

PONE-D-25-13867R1

PLOS ONE

Dear Dr. De,

I'm pleased to inform you that your manuscript has been deemed suitable for publication in PLOS ONE. Congratulations! Your manuscript is now being handed over to our production team.

Kind regards,

on behalf of

Dr. Vinod Kumar Vashistha

Academic Editor

PLOS ONE